# Question Decomposition using Masked Language Modeling for Knowledge Editing

## Abstract

Large Language Models (LLMs) acquire vast amounts of knowledge during computationally expensive pre-training. Knowledge Editing has emerged as a lightweight bypass for updating factual information in LLMs. To handle multi-hop question-answering (MQA), knowledge editors rely on compositional reasoning to decompose multi-hop questions into their constituent subquestions. State-of-the-art knowledge editors perform question decomposition using large causal language models, which often introduce errors manifested as hallucinations. In this paper, we propose Question Decomposition using Masked Language Modeling for Editing Knowledge (QMEK), a knowledge editing framework for multi-hop question-answering. The framework consists of two key components: a question decomposition module and a subquestion answering module. Our approach is motivated by the insight that reformulating question decomposition as a *masked language modeling* task rather than a *causal language modeling* task reduces inference complexity and curbs hallucinations. Furthermore, we adopt a relational triple representation in both modules to eliminate errors that arise when performing translations between natural language and structured triple formats. We evaluate QMEK against 5 state-of-the-art frameworks on 3 datasets and achieve an average 17.5% accuracy increase and 10.2× speedup.

## 1 Introduction

Large Language Models (LLMs) have demonstrated strong performance across a range of tasks, including question-answering (Du et al., 2025; Kamalloo et al., 2023), code generation (Gu et al., 2024a), and machine translation (Zhu et al., 2024). These models learn knowledge through a computationally intensive training process (Li et al., 2023; Maslej et al., 2025), making it impractical to re-train them to update factual information. To mitigate this limitation, knowledge editing provides a lightweight mechanism for revising LLMs' internal knowledge without retraining.

Knowledge editing frameworks typically rely on injecting updates to specific locations of an LLM's parameters and achieve considerable success in factual information updates (Hu et al., 2024; Tan et al., 2024). For example, editors can successfully modify LLMs such that *Linus Torvalds* is now the *creator* of *Windows* instead of *Linux*. When the model is queried with *What operating system did Linus Torvalds create?*, it now responds with *Windows* instead of *Linux*. However, these approaches fail to cause the model to recall edited knowledge when faced with questions requiring compositional reasoning (Zhong et al., 2023). Combining the above edit with the multi-hop question *What country is the creator of Windows a citizen of?* should cause the edited model to respond with *Finland*, but often the model will not recall the edit and instead responds with the pre-edit answer *United States*.

To reliably recall edits for use in compositional reasoning tasks such as multi-hop question-answering (MQA), knowledge editors began storing edited information in external structures (Gu et al., 2024b; Simon & Ewetz, 2025; Wang et al., 2024; Zhong et al., 2023). During inference, multi-hop questions are decomposed into subquestions using large causal language models (CLMs) such as GPT (Brown et al., 2020) and Llama (Grattafiori et al., 2024). Next, each subquestion is answered by querying the external structure for relevant edits. We observe that question decomposition is the bottleneck of state-of-the-art knowledge editors. Specifically, large CLMs that iteratively generate the next token often fail to correctly decompose multi-hop questions due to errors introduced by hallucinations. We speculate that this challenge could potentially be overcome by directly

identifying the subquestions within the original multi-hop question. Another challenge is that many recent editing frameworks decompose a question into structured triples and then translate them into natural-language subquestions. This triplet-to-text step introduces an extra failure mode, inviting hallucinations and compounding downstream errors.

In this paper, we present Question Decomposition using Masked Language Modeling for Editing Knowledge (QMEK), a knowledge editing framework for multi-hop question-answering (MQA). The proposed framework consists of a question decomposition module and a subquestion answering module. It builds on the insight that large CLMs are not well-suited for question decomposition tasks. Instead, it performs question decomposition using smaller language models such as BERT (Devlin et al., 2019) via masked language modeling. Moreover, it provides a unified triplet representation for capturing multi-hop questions to avoid translating questions between different internal representations. The main contributions of this paper can be summarized as follows:

- We reformulate multi-hop question decomposition into a masked language modeling task, which identifies subquestions within the original multi-hop question. This significantly reduces the task complexity compared with the original causal language modeling formulation, i.e., token classification vs. iterative token generation.

- We leverage a unified relational triple representation for both question decomposition and subquestion answering. The unified representation eliminates translation errors and enables subquestions to be answered through structured triplet completion. We also use a two-tier solution for identifying relevant edits in an edit bank.

- In our experimental evaluation, we compare QMEK with 5 state-of-the-art MQA-focused knowledge editors over 3 datasets. We observe that QMEK is on average $17.5\%$ more accurate and $10.2\times$ faster.

The remainder of the paper is organized as follows: the problem formulation and related works are presented in Section 2, the motivation for using a masked language model is given in Section 3, the methodology is explained in Section 4, experimental results are discussed in Section 5, and the paper is concluded in Section 6.

## 2 BACKGROUND

In this section, we first introduce the MQA problem formulation considered in this paper. Next, we review closely related work on knowledge editing.

### 2.1 PROBLEM FORMULATION

Knowledge editors formulate LLM knowledge as a set of factual associations, such as *Hideo Kojima was born in Japan*. Factual associations can be simplified to relational triples $t = (s, r, o)$, where subject entities $s$ and object entities $o$ are linked by some relation $r$. The above factual association can be represented as (*Kojima*, *born*, *Japan*).

Single-hop questions can also be formulated as a relational triple, allowing multi-hop questions to become sets of relational triples:

$$\mathcal{T} = \{(s, r_0, o_0), (o_0, r_1, o_1), ..., (o_{n-1}, r_n, o_n)\}, \tag{1}$$

where each question hop maps to a relational triple. The multi-hop question *What is the capital of the country of birth of Hideo Kojima?* becomes $\{(Kojima, born\ in, Japan), (Japan, capital, Tokyo)\}$. Relational triples are updated $t^* = (s, r, o^*)$ such that the subject $s$ is connected by the relation $r$ to a new object $o^*$. Multi-hop edits are be achieved by applying edits across intermediate question hops such that the triple set $\mathcal{T}$ becomes:

$$\mathcal{T}^* = \{(s, r_0, o_0^*), (o_0^*, r_1, o_1^*), ..., (o_{n-1}^*, r_n, o_n^*)\}. \tag{2}$$

Editing an object in one triple also modifies the subjects and objects of the downstream triples.

Answering multi-hop questions in the presence of edited knowledge is the purpose of MQA-focused knowledge editors. However, accurately decomposing these questions into their constituent question hops for effective editing is a challenging task which we address in this paper.

Table 1: Methods and representations for MQA. Question decomposition and subquestion answering are performed by a causal language model (CLM), a masked language model (MLM), or a knowledge graph (KG). Both tasks use natural language (NL), relational triples (triplet), or knowledge graph (KG) representations.

| Work in | Question Decomposition | | Subquestion Answering | |
|---|---|---|---|---|
| | Methods & Models | Representation | Methods & Models | Representation |
| MeLLo | CLM | NL | CLM | NL |
| DeepEdit | CLM | NL | CLM | NL |
| PokeMQA | CLM | NL | CLM | NL |
| RAE | CLM | NL | CLM + KG | KG |
| GMeLLo | CLM | triplet | CLM + KG | KG |
| CHECK | CLM | triplet | CLM | NL |
| QMEK (ours) | MLM | triplet | CLM | triplet |

## 2.2 RELATED WORKS

In this section, we provide an overview of related work. We center our discussion on the models and data representations used by knowledge editors during question decomposition and subquestion answering, visualized in Table 1.

**Question Decomposition:** State-of-the-art knowledge editors perform question decomposition using causal language models (CLMs) such as GPT and Llama, typically by reformulating a multi-hop question into a sequence of subquestions. These subquestions can be represented either in natural language (Yu et al., 2024; Wang et al., 2024; Gu et al., 2024b; Shi et al., 2024) or in a structured triplet format (Chen et al., 2024; Simon & Ewetz, 2025). The triplet format captures the decomposition using a *relationship chain* $\mathcal{C} = (s, r_0, \cdots, r_n)$, which includes the subject $s$ and relations $r_0$ to $r_n$ of the original multi-hop question. This format has proven to be more effective, as it enables the entire decomposition to be generated in a single pass. In contrast, natural language decomposition often proceeds iteratively, requiring the answer to each subquestion before generating the next. This step-by-step approach is prone to cascading errors due to unintended rephrasings. Rather than relying on CLMs for the triplet extraction, we decompose the multi-hop question into subquestions with a masked language model, yielding cleaner triplets and fewer hallucinations.

**Subquestion Answering:** Decomposed subquestions are answered using a CLM (Chen et al., 2024; Gu et al., 2024b; Wang et al., 2024; Simon & Ewetz, 2025; Zhong et al., 2023) which may be supplemented with a knowledge graph in some frameworks (Cheng et al., 2024; Shi et al., 2024). Knowledge graph editors modify knowledge graph representations with edited information and traverse the graphs to answer questions. Graph-based editors are promising for domain-specific applications, but are unable to function in the absence of knowledge graphs. Answering subquestions using a CLM is straightforward when the decomposed question is represented in natural language. An edit bank implemented using vector embeddings is used to check each subquestion for edits. However, if the decomposed question is represented using a relationship chain $\mathcal{C}$, the current subquestion must be translated back into natural language. For example, (*Japan*, *capital*, *?*) is required to be translated into *What is the capital of Japan?*, which may introduce translation errors. Our proposed solution circumvents these translation errors by directly answering subquestions through triplet completion.

## 3 MOTIVATION

In this section, we analyze the capability of state-of-the-art knowledge editors at decomposing multi-hop questions into subquestions. The subquestion decomposition is necessary in order to check if any of the intermediate reasoning steps have been subject to an edit. Previous editors use CLMs to extract question hops as natural language subquestions (Gu et al., 2024b; Wang et al., 2024; Zhong et al., 2023) or relational triples (Chen et al., 2024; Simon & Ewetz, 2025).

We compare the two approaches using the representative frameworks PokeMQA (NL) and CHECK (triplets) in Figure 2. The frameworks are evaluated on the ability to decompose questions into the correct number of hops on the MQuAKE-2002 (Wang et al., 2024) dataset using Llama 3.1 (Grattafiori et al., 2024). The figure shows that question decomposition using natural language achieves a decomposition accuracy of 7%, 33%, 53% for

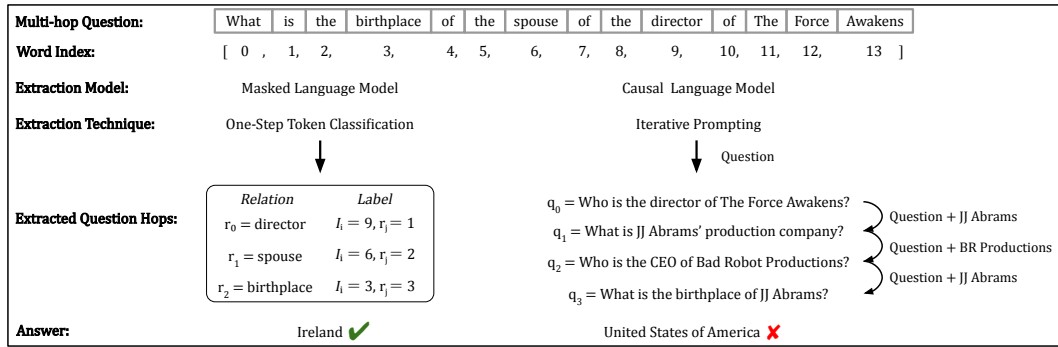

Figure 1: A multi-hop question decomposed using a masked (left) and causal (right) language model. The masked language model labels relations in text and directly extracts the labeled relations to represent question hops. The causal model iteratively generates new subquestions, with previous subquestion answers being used as context for the next subquestion.

2-, 3-, and 4-hop questions, respectively. The triplet representation improves the decomposition accuracy to 65%, 65%, and 54%, respectively. While this is a significant improvement, it is important to note that the question decomposition accuracy acts as a hard upper bound on the question-answering accuracy prior to subquestion answering occurring.

With peak decomposition accuracies at 65%, it is impossible to answer 35% of the decomposed questions. Such decomposition accuracies are unacceptable if the objective is to attain overall accuracies in the 90% range. This gap motivated us to rethink how to perform question decomposition for knowledge editing.

We contrast our proposed approach based on masked language modeling with the state-of-the-art approach based on causal language modeling in Figure 1. CLMs iteratively generate a single next token prediction for some input token sequence. While CLMs have remarkable generation capabilities, they are required to generate

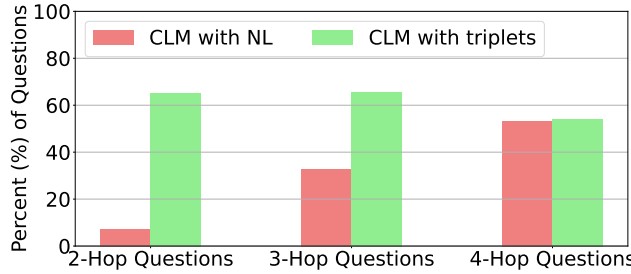

Figure 2: A comparison of question decomposition for two frameworks that use causal language modeling (CLM) to decompose questions into natural language (NL) or relational triples (triplets). Evaluations are performed on the MQuAKE-2002 dataset.

complete subquestions or relational triples based on context from the original multi-hop question. This may introduce errors by (i) incorrectly paraphrasing subquestions and (ii) by incorrectly identifying the number of hops in the question, shown in Figure 1 (right). We speculate that these challenges may stem from the model needing to select one option from its vocabulary of tens of thousands of options for each generated token. In contrast, we observe that the relations that we aim to extract are present in the original multi-hop question. Therefore, there is an opportunity to formulate relation extraction as a masked language modeling task, where the problem is to identify whether each word is part of a relation or not. This simplifies relation extraction to a classification problem with $n_{max}$ options, where $n_{max}$ is the maximum number of question hops.

## 4 METHODOLOGY

In this section, we present the methodology of the QMEK framework. The input consists of a set of edits and a multi-hop question, and the output is the final answer to that question. Before answering, the edits are embedded into an edit bank. As illustrated in Figure 3, the framework performs multi-hop question-answering (MQA) in two stages: question decomposition and subquestion answering. In the first stage, the framework decomposes the multi-hop question into single-hop subquestions

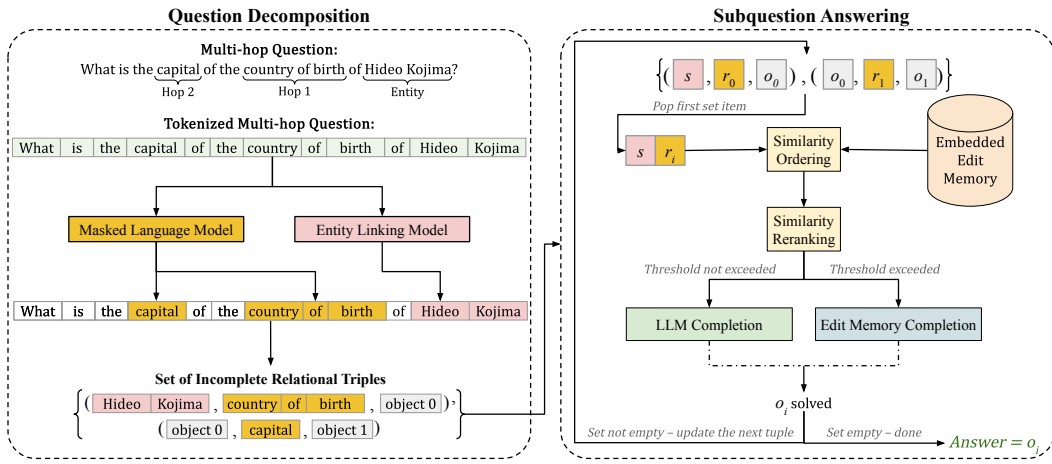

Figure 3: An overview of the QMEK framework. **Question Decomposition**: A multi-hop question is decomposed into the question entity and relations using an entity linking model and masked language model. The extracted components are combined to form a set of incomplete triples. **Subquestion Answering**: The subject and relation of each triple is compared against all edits using cosine similarity and a cross-encoder. Triples are completed with LLM knowledge or an edit.

using masked language modeling. Each subquestion is represented as an incomplete relational triple, as detailed in Section 4.1. In the second stage, each subquestion is answered sequentially via relational triple completion. Before answering, the framework checks whether the subquestion has been modified by any of the edits in the edit bank. The details of this process are provided in Section 4.2.

## 4.1 QUESTION DECOMPOSITION

In this section, we describe the question decomposition module. The module takes in a multi-hop question and outputs a set of incomplete relational triples:

$$T^u = \{(s, r_0, o_0^u), (o_0^u, r_1, o_1^u), ..., (o_{n-1}^u, r_n, o_n^u\}, \tag{3}$$

where $o_n^u$ represents an unknown entity. The subquestions answered in the next section are directly defined by the incomplete triples. We decompose the problem of determining $T^u$ into a subject extraction problem (identifying $s$) and a relation extraction problem (identifying $r_0, \ldots, r_n$).

**Subject Extraction using Entity Linking Models:** Subject extraction aims to identify the subject $s$ of the multi-hop question. Entity linking models are pre-trained to locate entities in text and link extracted entities to a known entity from the knowledge base the model was trained on. We use the ReFinED (Ayoola et al., 2022) entity linking model to identify the subject of each multi-hop question. Subject extraction is completed by first passing the multi-hop question $\mathcal{Q}$ to the entity linking model $f_e$. The entity linking model $f_e$ then extracts a set of entities $\mathcal{S} = \{s_0, s_1, ..., s_n\} = f_e(\mathcal{Q})$. We discard any numeric extractions because we are only interested in relations between entities. We consider the final extracted entity $s_n \in \mathcal{S}$ to be the question entity because the entities of interest are typically at the end of the multi-hop question.

**Relation Extraction using Masked Language Modeling:** Relation extraction aims to identify the relations within the multi-hop question $Q$ that represent question hops. The relation extraction in QMEK is performed by fine-tuning an MLM to assign an indicator variable $\mathcal{I}$ to each token of the input. The indicator variable is in the form of an integer. The integer $\mathcal{I}_i$ specifies if token $i$ is part of a relation, as follows:

$$\mathcal{I}_i = \begin{cases} 0, & \mathcal{I}_i \text{ is not part of a relation,} \\ j, & \mathcal{I}_i \text{ is part of } r_j \text{ with } j \in [1, \text{len}(\mathcal{Q})] \end{cases} \tag{4}$$

where $j$ is the relation id and $len(\cdot)$ describes the number of question hops in a multi-hop question $\mathcal{Q}$. The masked language modeling formulation and notation is show in Figure 1 (left). By assigning

each of the words to a relation, accidental paraphrasing is circumvented and the model extracts the correct number of hops with a higher accuracy.

We use a BERT (Devlin et al., 2019) model as our MLM. The model takes a tokenized question as input and outputs a 1-by-$n_{max}$ logit array for each input token, where $n_{max}$ is the maximum number of allowed question hops. All available datasets use between 1 and 4 question hops and a *no relation* label is necessary, so we use $n_{max} = 5$. The logit labels are described in Equation 4. We generate a finetuning dataset, described in Appendix Section G, derived from MQuAKE Zhong et al. (2023) that does not contain any overlapping edit cases with the datasets used during framework evaluations. The dataset is composed of edit cases, with each case containing between 1 and 4 questions that are paraphrases of each other. Each case also contains the set of question hops that represent the case questions. The finetuning inputs are the questions in the training dataset edit cases. The ground truths are one-hot encoded labels representing the relation $j$ each input token belongs to, generated from the training dataset. Finetuning is achieved through gradient descent, by comparing the model output logits and ground truth labels using cross-entropy loss (Mao et al., 2023) with the AdamW (Loshchilov & Hutter, 2019) optimizer that has a constant learning rate of $1e^{-4}$. The model in use is pre-trained, so only 2 epochs are required for the model to converge. We used a batch size of 32 questions.

## 4.2 SUBQUESTION ANSWERING

In this section we describe the subquestion answering module. The input to the module is the set of incomplete relational triples $\mathcal{T}^u$, which are sequentially completed. By finding the unknown object $o_n^u$ for some triple $t_n^u$, we also find the subject $o_{n+1}$ of the next triple $t_{n+1}^u$. This allows for iterative triple completion until all incomplete triples are completed with an object. The output of the module is the object of the final triple, which is the answer to the multi-hop question.

The method for triple completion is determined by a two-round thresholding approach. The first round relies on cosine similarity thresholding. The second round relies on cross-encoder similarity thresholding. If both similarity thresholds are exceeded, the edit memory provides the new object for $t_n^u$. If either of the similarity thresholds are not exceeded, the LLM is used to complete the triple.

**Edit Storage:** Prior to question-answering, all edits are inserted into an embedding space. Edits are typically provided as cloze-style sentences. However, the edits must be converted to subject, relation pairs $(s, r)$ to match the format of the known subjects and relations of the incomplete triple set $\mathcal{T}^u$. First, all common English stopwords (Loper & Bird, 2002) are removed from the edits. Next, all capitalized words are moved to the beginning of the pruned edits. Finally the set of pruned edits $\mathcal{E}$ is inserted into the embedding space $\omega$ using a dense retrieval model $f_\omega$ such that $\mathcal{E}^\omega = f_\omega(\mathcal{E})$. QMEK uses the Qwen 3 embedding model (Zhang et al., 2025). The edit objects $o^\mathcal{E}$ are stored as a set of strings $\mathcal{O}^\mathcal{E} = \{o_0^\mathcal{E}, o_1^\mathcal{E}, ..., o_n^\mathcal{E}\}$.

**Triple Similarity Thresholding:** Each of the incomplete relational triples $t^u \in \mathcal{T}^u$ are sequentially completed by finding the missing object $o_n^u$, starting with the first triple $t_0^u = (s, r_0, o_1^u)$. The current triple $t_n^u$ goes through two rounds of similarity thresholding. We define the variable $\rho$ to track whether the current triple $t_n^u$ has passed both rounds of thresholding. The thresholding process can then be described as:

$$\rho = \begin{cases} 1, & (cos(t_n^u, e) > \tau - \lambda) \wedge (f_{CE}(t_n^u, e) > \tau), \\ 0, & (cos(t_n^u, e) \leq \tau - \lambda) \vee (f_{CE}(t_n^u, e) \leq \tau). \end{cases} \tag{5}$$

The first round of thresholding is comparing the current triple $t_n^u$ against all of the edits using cosine similarity. The subject $o_n^c$ and relation $r_{n+1}^c$ of the current triple $t_n^u$ are joined together to form a subject, relation pair $p = (o_n^c, r_{n+1}^c)$. The pair are then inserted into the embedding space $p^\omega = f_\omega(p)$ using the dense retrieval model $f_\omega$. Next, the embedded pair $p^\omega$ is compared against each edit $e^\omega \in \mathcal{E}^\omega$ using cosine similarity $cos(\cdot, \cdot)$. If any edits exceed the cosine similarity threshold $\tau - \lambda$, $\rho$ is set to 1 and the next round of similarity thresholding begins. If no edits exceed the threshold, $\rho$ is set to 0 and similarity thresholding ends.

The second round of similarity thresholding employs a cross-encoder to eliminate errors stemming from cosine similarities near the similarity threshold $\tau - \lambda$. Cross-encoders are commonly used to rerank the top-$k$ documents received from a dense retrieval model, taking as input string pairs and outputting a similarity score between the strings on the range $[0.0, 1.0]$. QMEK uses the Modern-

BERT (Reimers & Gurevych, 2019) cross-encoder. The subject, relation pair string $p$ is matched with each top-$k$ cosine similarity edit string $e$ and passed to the cross-encoder $f_{CE}(\cdot, \cdot)$. If the cross-encoder $f_{CE}(\cdot, \cdot)$ assigns a similarity score exceeding the threshold $\tau$, $\rho$ continues to be set to 1. If no edits exceed the threshold, $\rho$ is set to 0. It should be noted that the cosine similarity threshold $\tau - \lambda$ is the cross-encoder threshold $\tau$ with an offset to ensure that edits with similarities falling below $\tau$ are also considered for reranking, in addition to those that exceed $\tau$. The results of parameter searches for the optimal $k$, $\tau$, and $\lambda$ are provided in Appendix Section F.

**Edit Memory Completion:** If $\rho$ is 1 after both rounds of thresholding, the current triple is completed using information from the edit memory. The edit with the greatest similarity assigned by the cross-encoder $e_{relevant} = \max(f_{CE}(p, e))$ is the most relevant edit to the current hop, so the corresponding object $o^{\mathcal{E}}$ is retrieved from the set of edit objects $\mathcal{O}^{\mathcal{E}}$ and used to complete the current triple.

**LLM Triple Completion:** An edit is unnecessary if $\rho$ is ever set to 0 and LLM knowledge is used to complete the triple. Previous frameworks formulate LLM completion as a natural language question-answering task. However, that formulation requires translating the subject, relation pair $p$ into a question using a CLM. Given that the current question hop is formatted as a relational triple, we query the model with a relational triple completion task. This reduces error to hallucinations from translation, and reduces token usage and runtime. The subject, relation pair $p$ is passed to the LLM $\mathcal{F}$ with an in-context learning prompt $\mathcal{P}$ for triple completion $o^{\mathcal{F}} = \mathcal{F}(p|\mathcal{P})$. The in-context learning prompt $\mathcal{P}$ is described in Appendix Section H.

## 5 EXPERIMENTAL RESULTS

**Baselines:** We compare QMEK against 4 other vector embedding editors: MeLLo (Zhong et al., 2023), DeepEdit (Wang et al., 2024), PokeMQA (Gu et al., 2024b), and CHECK (Simon & Ewetz, 2025). We also compare against the question-answering portion of GMeLLo (Chen et al., 2024) for comparison against a RAG-like (Lewis et al., 2020) framework. Experimental parameters used by the frameworks are provided in Appendix Section B.1

**Models and Datasets:** Evaluations are performed using 3 state-of-the-art open-source LLMs and 4 commonly used datasets. The models are Qwen 2.5 7B (Yang et al., 2025), Mistral v0.3 7B (Jiang et al., 2023), and Llama 3.1 8B (Grattafiori et al., 2024). The datasets are MQuAKE-Remastered (Zhong et al., 2025), MQuAKE-2002 (Wang et al., 2024), MQuAKE-Hard (Wang et al., 2024), and KEBench (Wei et al., 2024).

**Metrics:** The MQuAKE dataset is broken into sets of multi-hop questions referred to as cases. Each case is composed of 3 multi-hop questions that contain the same content, but rephrased. A case is considered correct if at least one of its multi-hop questions was answer correctly. Case accuracy is defined as $\frac{case_{correct}}{case_{total}} \times 100$. A question is considered correct if the denoted answer is the same as the ground truth answer or a ground truth answer alias. Question accuracy is defined as $\frac{question_{correct}}{question_{total}} \times 100$. We also evaluate frameworks on decomposition accuracy. A question is correctly decomposed if the number of extracted questions hops equals the true number of question hops. Decomposition accuracy is defined as $\frac{decomp_{correct}}{decomp_{total}} \times 100$.

### 5.1 QUESTION-ANSWERING ACCURACY

An evaluation of QMEK and other state-of-the-art Knowledge Editing frameworks on question-answering accuracy is provided in Table 2. The frameworks are evaluated across the MQuAKE datasets and all models described above. Across all datasets and LLMs, QMEK achieves the highest per-case and per-question accuracies, with an average 15.33% and 17.54% increase over the next highest accuracy, respectively. A notable performance improvement is the average 29.91% and 27.01% per-case and per-question increase over the next highest accuracies on the MQuAKE-Hard dataset, which is comprised of only 4-hop questions that have 3 or 4 question hops requiring editing. Overall, these performance gains demonstrate that QMEK can correctly answer more unique edit scenarios and more questions per edit scenario due to the superior question decomposition approach. Question-answering evaluations on additional models and datasets are provided in Appendix Section C. Ablation studies over question hops and edits are provided in Appendix Section D.2.

Table 2: Per-case and per-question accuracy across the MQuAKE datasets. Top accuracy per column and per model is bolded.

| Dataset (Zhong et al., 2023) | MQuAKE-R | | MQuAKE-2002 | | MQuAKE-Hard | |
|---|---|---|---|---|---|---|
| Accuracy Type (%) | Case | Question | Case | Question | Case | Question |
| Model | Qwen 2.5 (Yang et al., 2025) | | | | Size: 7B | |
| GMeLLo-QA (Chen et al., 2024) | 13.24 | 8.24 | 14.09 | 8.37 | 7.69 | 3.34 |
| MeLLo (Zhong et al., 2023) | 23.97 | 11.94 | 33.22 | 16.67 | 32.17 | 17.33 |
| DeepEdit (Wang et al., 2024) | 33.20 | 20.20 | 50.65 | 38.44 | 3.26 | 1.55 |
| PokeMQA (Gu et al., 2024b) | 43.47 | 26.37 | 44.11 | 25.22 | 20.75 | 9.87 |
| CHECK (Simon & Ewetz, 2025) | 44.60 | 28.19 | 63.54 | 41.84 | 46.39 | 28.75 |
| QMEK (ours) | **55.90** | **46.08** | **73.48** | **57.59** | **76.92** | **58.35** |
| Model | Mistral v0.3 (Jiang et al., 2023) | | | | Size: 7B | |
| GMeLLo-QA (Chen et al., 2024) | 13.03 | 8.13 | 10.59 | 7.01 | 4.66 | 2.41 |
| MeLLo (Zhong et al., 2023) | 33.20 | 20.20 | 34.12 | 21.30 | 7.69 | 3.50 |
| DeepEdit (Wang et al., 2024) | 36.10 | 29.79 | 49.80 | 39.16 | 1.63 | 3.26 |
| PokeMQA (Gu et al., 2024b) | 44.40 | 32.34 | 55.54 | 37.80 | 33.80 | 21.06 |
| CHECK (Simon & Ewetz, 2025) | 50.67 | 35.48 | 66.13 | 46.59 | 45.92 | 32.25 |
| QMEK (ours) | **56.93** | **46.73** | **73.88** | **57.99** | **76.69** | **58.12** |
| Model | Llama 3.1 (Grattafiori et al., 2024) | | | | Size: 8B | |
| GMeLLo-QA (Chen et al., 2024) | 12.60 | 6.63 | 9.24 | 4.83 | 6.29 | 2.95 |
| MeLLo (Zhong et al., 2023) | 30.30 | 14.31 | 28.12 | 13.32 | 9.79 | 3.81 |
| DeepEdit (Wang et al., 2024) | 39.37 | 25.50 | 50.85 | 32.33 | 2.80 | 1.09 |
| PokeMQA (Gu et al., 2024b) | 45.07 | 26.89 | 53.75 | 30.59 | 34.97 | 17.72 |
| CHECK (Simon & Ewetz, 2025) | 51.30 | 38.81 | 67.68 | 47.07 | 48.25 | 32.71 |
| QMEK (ours) | **57.03** | **47.13** | **74.98** | **59.34** | **76.69** | **58.28** |

## 5.2 QUESTION DECOMPOSITION ACCURACY

We present an evaluation on the question decomposition accuracy of QMEK and other frameworks on the MQuAKE-2002 dataset in Figure 4(a). PokeMQA achieves a low decomposition accuracy of 30% by using a CLM to extract natural language question hops. CHECK decomposition improves upon PokeMQA decomposition by using a relational triple representation rather than natural language and achieves a 51% accuracy. However, CLMs are prone to hallucinations and so extract hallucinated relations. QMEK eliminates many hallucinations by extracting relations with an MLM and achieves the highest decomposition accuracy of 74%. Clearly, there is still room to improve question decomposition, but these results indicate that abandoning CLMs in favor of models with other objectives results in better question decomposition.

## 5.3 EVALUATION ON UNSEEN DATA

While the MLM used by QMEK is trained on a subset of the MQuAKE dataset not used in any of the evaluations, all questions in the MQuAKE subsets share strong sentence structure similarities. Therefore, it is necessary to evaluate the performance of the MLM on an unseen distribution of data. To this end, we provide an evaluation of QMEK and other editors on the KEBench dataset in Figure 4(b). PokeMQA and CHECK achieve average decomposition accuracies of 4% and 52%, respectively. QMEK achieves a decomposition accuracy of 93% on the KEBench dataset, a 36% accuracy increase over CHECK. This high decomposition accuracy again demonstrates the effectiveness of masked language modeling for question decomposition. Furthermore, it provides evidence towards the ability of masked language modeling finetuned for question decomposition to generalize to unseen data. We also provide question-answering accuracy evaluations on KEBench in Appendix Section C.2. Finally, to showcase the ability of QMEK's MLM to generalize to unseen relations, we provide question decomposition evaluations on the RippleEdit Cohen et al. (2024) dataset in Appendix Section C.3.

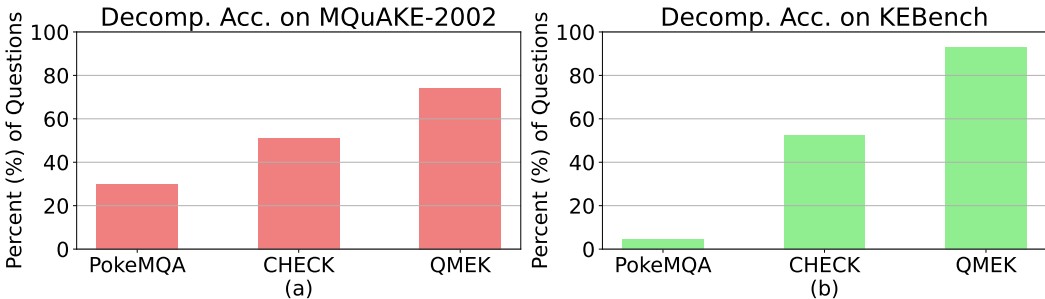

Figure 4: PokeMQA and CHECK use the Llama 3.1 LLM while QMEK uses the finetuned MLM for question decomposition. (a) Question decomposition accuracy on the MQuAKE-2002 dataset. (b) Question decomposition accuracy over the KEBench dataset.

Table 3: Per-question runtime in seconds across the MQuAKE datasets. The highest accuracy per column and per model is bolded.

| Dataset | | MQuAKE-2002 | | | MQuAKE-Hard | | |
|---|---|---|---|---|---|---|---|
| Model | | Qwen 2.5 | Mistral v0.3 | Llama 3.1 | Qwen 2.5 | Mistral v0.3 | Llama 3.1 |
| MeLLo | | 3.47 | 5.28 | 7.34 | 4.30 | 5.92 | 8.13 |
| DeepEdit | | 4.53 | 5.83 | 14.58 | 5.08 | 6.88 | 17.52 |
| PokeMQA | | 2.10 | 2.82 | 5.10 | 2.28 | 3.13 | 5.81 |
| CHECK | | 2.82 | 2.53 | 4.96 | 2.39 | 2.29 | 4.82 |
| QMEK | | **0.20** | **0.23** | **0.46** | **0.23** | **0.27** | **0.46** |

## 5.4 RUNTIME EVALUATION

Knowledge editors are meant to be computationally efficient methods of parameter modification when updating model knowledge is necessary. One measure of editing efficiency is the average runtime required for an LLM modified with an editor to answer a question. We present the average per-question runtime of QMEK and other vector embedding frameworks in Table 3. The runtimes are provided for the MQuAKE-2002 and MQuAKE-Hard datasets the previously described LLMs. MeLLo, PokeMQA, DeepEdit, and CHECK take an average 5.74, 3.54, 9.07, and 3.30 seconds. QMEK only requires an average 0.31 seconds, making it $10.2\times$ faster than the next fastest framework. We attribute the large runtime decrease to QMEK using only a single LLM model call per question hop. Model calls are the most computationally expensive components of the vector embedding frameworks. In contrast, methods that rely on CLMs make at least one model call for question decomposition and at least two model calls for subquestion-answering, leading to inflated runtimes. Runtime ablation studies over the number of question hops and edits are provided in Appendix Section D.3.

## 6 CONCLUSION

In this paper, we introduce the QMEK Knowledge Editing framework. Previous editors experience a performance bottleneck at the question decomposition stage of multi-hop question-answering due to a reliance on causal language models. We show that causal language models often incorrectly decompose questions due to hallucinations. To avoid hallucinations and improve question decomposition, we propose using masked language models to extract relations directly from questions. The direct extraction of question relations allows QMEK to formulate subquestion answering as a relational triple completion task rather than a natural language question-answering task. Subquestion answering is then completed using a two-stage thresholding approach for embedding and string similarity between each intermediate reasoning step and all edits. Our evaluations show the QMEK achieves a $17.5\%$ average accuracy increase over the next highest editor and a $10.2\times$ faster runtime.

## REPRODUCIBILITY STATEMENT

We provide the code for QMEK and the code for generating experimental results with QMEK in the supplementary materials. We will also release a Github Repository containing this code, contingent on the paper's acceptance.

## ETHICS STATEMENT

The purpose of the proposed work is to reduce the computational, financial, and environmental burden of updating Large Language Models (LLMs). There are no ethical considerations unique to this work. As with all work on improving LLMs, there is always the concern for the dissemination of incorrect information by these models. Additionally, LLMs were used to aid in the polish of text. Specifically, LLMs were used to check for grammatical consistency and rephrase text for clarity.

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

## A  OVERVIEW

We provide an additional 7 sections in the Appendix. The different baseline frameworks, datasets, and Large Language Models are discussed in Section B. Additional question decomposition and question-answering evaluations are provided in Section C. Ablations studies over per-hop and per-edit accuracy are provided in Section D. A study on token utilization of QMEK and other frameworks is provided in Section E. Parameter searches to find the optimal parameters used by QMEK are provided in Section F. How to generate the dataset used to train QMEK's masked language model is explained in Section G. The in-context learning prompt used for the relational triple completion task is given in Section H.

## B    Baselines, Datasets, and Models Discussion

In this section, we discuss the frameworks used as comparisons, the datasets evaluated over, and the Large Language Models (LLMs) used in evaluations. Other frameworks are discussed in Section B.1. The datasets used in evaluations are covered in Section B.2. The models used in evaluations are discussed in Section B.3.

### B.1    Vector Embedding Frameworks

We use 4 other vector embedding frameworks to put the performance of QMEK in context. These frameworks are MeLLo (Zhong et al., 2023), DeepEdit (Wang et al., 2024), PokeMQA (Gu et al., 2024b), and CHECK (Simon & Ewetz, 2025). These are the only vector embedding frameworks that we know of. We also use GMeLLo (Chen et al., 2024) as a baseline. GMeLLo has a RAG (Lewis et al., 2020) component and a knowledge graph component. We only compare against the RAG component.

We set the *maximum new tokens*, *maximum number of extracted question hops*, and *cosine similarity threshold*, *cross-encoder threshold*, *cross-encoder threshold offset*, and *cross-encoder top-k* framework parameters prior to evaluations. We set the *maximum new tokens* generated from each model call to 200 because we found that the prompts used by frameworks typically output a number of tokens far under that amount. We set the *maximum number of extracted question hops* to be 5. Through analysis of framework output logs for frameworks that iteratively generate subquestions, we found that frameworks iteratively generating more than 5 subquestions tend to start repeating a single subquestion, never arriving at an answer. By cutting those frameworks off after 5 subquestions generated, we shorten the time it takes to complete evaluations without harming framework performance. CHECK uses a *cosine similarity threshold* of 0.8, which was provided as the best performing threshold in the paper. QMEK uses a *cross-encoder threshold* $\tau$ of 0.7, a *cross-encoder threshold offset* $\lambda$ of 0.05, and a *cross-encoder top-k* of $k = 20$. Parameter searches for the best parameter values used by QMEK are provided in Section F.

For the sake of reproducibility, we now provide the hardware used to run all experiments. All experiments were run on a computing cluster using 1 NVIDIA B200, 16 CPU cores, and 64 GB of RAM, Red Hat Enterprise Linux version 9.5, and Python version 3.9.21.

### B.2    Datasets

We perform evaluations over the MQuAKE-CounterFact-3k and MQuAKE-Temporal (Zhong et al., 2023) datasets. MQuAKE-CounterFact-3k contains 1000 2-hop, 3-hop, and 4-hop edit cases, for a total of 3000 edit cases. MQuAKE-Temporal contains 1421 2-hop, 445 3-hop, and 2 4-hop edit cases. Each edit case is broken into 3 questions, with all 3 questions being paraphrases containing the same question hops. The number of edits per edit case varies. These are the most commonly used knowledge editing for multi-hop question-answering datasets.

We provide evaluations on the MQuAKE-Remastered Zhong et al. (2025) dataset. The dataset was created using the same methods as MQuAKE-CounterFact and MQuAKE-Temporal, but with additional safeguards to ensure that edit cases do not contain any conflicting edits. The dataset contains a total of 9171 edit cases with 6334 unique edits. We shorten the dataset to 1000 of each 2-hop, 3-hop, and 4-hop edit cases for a total of 3000 edit cases.

We provide evaluations over MQuAKE-2002 and MQuAKE-Hard (Wang et al., 2024), which are modifications of the MQuAKE-CounterFact-3k dataset. MQuAKE-2002 removes 998 edit cases that contain edits that conflict with other edit cases. MQuAKE-Hard contains 429 4-hop edit cases with each case containing 4 edits. We use MQuAKE-2002 because it is a more consistent dataset than MQuAKE-CounterFact-3k and MQuAKE-Hard for its more challenging edit cases.

Finally, we perform evaluations over KEBench (Wei et al., 2024) and Ripple Edit (Cohen et al., 2024). KEBench contains 2798 2-hop questions, each containing only 1 edit. Ripple Edit is composed of 3 subsets with different themes. Our evaluations cover 1201, 2243, and 5433 questions from the popular, random, and recent subsets, respectively. Both datasets are used to evaluate QMEK over unseen data.

Table 4: Per-case and per-question accuracy across the MQuAKE datasets. Top accuracy per column and per model is bolded.

| Dataset (Zhong et al., 2023) | MQuAKE-CF-3k | | MQuAKE-T | |
|---|---|---|---|---|
| Accuracy Type (%) | Case | Question | Case | Question |
| Model | Qwen 2.5 (Yang et al., 2025) | | Size: 7B | |
| GMeLLo-QA (Chen et al., 2024) | 13.90 | 7.99 | 47.91 | 28.10 |
| MeLLo (Zhong et al., 2023) | 26.73 | 13.17 | 62.69 | 33.62 |
| DeepEdit (Wang et al., 2024) | 35.53 | 26.47 | 80.62 | **66.18** |
| PokeMQA (Gu et al., 2024b) | 27.90 | 14.30 | 68.74 | 40.60 |
| CHECK (Simon & Ewetz, 2025) | 47.43 | 30.49 | 81.58 | 61.76 |
| QMEK (ours) | **51.33** | **39.72** | **81.80** | 60.92 |
| Model | Mistral v0.3 (Jiang et al., 2023) | | Size: 7B | |
| GMeLLo-QA (Chen et al., 2024) | 12.83 | 8.08 | 55.51 | 34.51 |
| MeLLo (Zhong et al., 2023) | 30.70 | 18.36 | 56.53 | 33.17 |
| DeepEdit (Wang et al., 2024) | 35.77 | 27.27 | 82.49 | **69.09** |
| PokeMQA (Gu et al., 2024b) | 42.50 | 27.68 | 79.50 | 53.73 |
| CHECK (Simon & Ewetz, 2025) | 48.00 | 33.30 | **86.67** | 60.76 |
| QMEK (ours) | **51.30** | **39.88** | 86.56 | 64.24 |
| Model | Llama 3.1 (Grattafiori et al., 2024) | | Size: 8B | |
| GMeLLo-QA (Chen et al., 2024) | 11.03 | 5.71 | 60.44 | 35.14 |
| MeLLo (Zhong et al., 2023) | 28.90 | 13.30 | 33.17 | 61.19 |
| DeepEdit (Wang et al., 2024) | 37.07 | 22.46 | 82.12 | 63.60 |
| PokeMQA (Gu et al., 2024b) | 37.00 | 18.33 | 56.00 | 26.07 |
| CHECK (Simon & Ewetz, 2025) | 50.57 | 34.63 | **85.87** | **64.61** |
| QMEK (ours) | **52.17** | **40.80** | 85.76 | 64.22 |

## B.3 MODELS

We provide evaluations using the Qwen 2.5 (Yang et al., 2025), Mistral v0.3 (Jiang et al., 2023), and Llama 3.1 (Grattafiori et al., 2024) LLMs. We use these LLMs because they are are recent and have strong language modeling capabilities. We also provide evaluations over GPT-J (Wang & Komatsuzaki, 2021), Vicuna (Chiang et al., 2023), and Falcon (Almazrouei et al., 2023) for easier comparison against results provided in previous works. All models contain between 6 and 8 billion parameters so that GPU constraints and forward pass runtimes are not issues.

## C ADDITIONAL DATASET EVALUATIONS

We provide additional question-answering accuracy on the MQuAKE (Wang et al., 2024; Zhong et al., 2023) datasets in Section C.1. We provide question-answering accuracy on the KEBench (Wei et al., 2024) dataset in Section C.2. We provide the decomposition accuracy of QMEK on the Ripple Edit (Cohen et al., 2024) dataset in Section C.3.

## C.1 MQUAKE

**Additional Datasets:** We provide results on MQuAKE-CounterFact-3k and MQuAKE-Temporal Zhong et al. (2023) in Table 4. These datasets have been used as the main modes of evaluation in previous works, but have been found to contain flaws, such as overlapping edits and questions that are not correctly structured. As a result, the current value of including these datasets is dubious, but for the sake of comparison with previous works, we choose to include them.

QMEK achieves the best performance on MQuAKE-CF-3k across all Large Language Models (LLMs). However, it consistently performs the second best across the language models on the MQuAKE-Temporal dataset by an average of $0.11\%$ per-case accuracy and $3.5\%$ per-question ac-

Table 5: Per-case and per-question accuracy across the MQuAKE datasets. The top accuracy per column and per model is in bold. Results reported in CHECK are denoted with *.

| Dataset (Zhong et al., 2023) | MQuAKE-CF-3k | | MQuAKE-T | |
|---|---|---|---|---|
| Accuracy Type (%) | Case | Question | Case | Question |
| Model | GPT-J (Wang & Komatsuzaki, 2021) | | Size: 6B | |
| GMeLLo-QA* (Chen et al., 2024) | 10.60 | 6.04 | 21.95 | 10.67 |
| MeLLo* (Zhong et al., 2023) | 14.97 | 6.89 | 32.82 | 18.49 |
| DeepEdit* (Wang et al., 2024) | 19.03 | 13.44 | 55.84 | 41.86 |
| PokeMQA* (Gu et al., 2024b) | 15.70 | 6.97 | 59.37 | 31.00 |
| CHECK* (Simon & Ewetz, 2025) | 42.27 | 29.57 | **78.69** | 55.82 |
| QMEK (ours) | **49.07** | **38.33** | 78.43 | **58.92** |
| Model | Vicuna (Chiang et al., 2023) | | Size: 7B | |
| GMeLLo-QA* (Chen et al., 2024) | 11.23 | 6.44 | 28.53 | 14.38 |
| MeLLo* (Zhong et al., 2023) | 9.93 | 5.08 | 68.52 | 50.18 |
| DeepEdit* (Wang et al., 2024) | 13.87 | 8.38 | 34.05 | 19.04 |
| PokeMQA* (Gu et al., 2024b) | 30.97 | 18.18 | 68.68 | 48.11 |
| CHECK* (Simon & Ewetz, 2025) | 47.57 | 30.93 | 81.64 | 55.84 |
| QMEK (ours) | **51.33** | **39.76** | **85.92** | **64.88** |
| Model | Falcon (Almazrouei et al., 2023) | | Size: 7B | |
| GMeLLo-QA* (Chen et al., 2024) | 7.77 | 4.27 | 16.38 | 7.57 |
| MeLLo* (Zhong et al., 2023) | 4.01 | 7.30 | 52.94 | 36.42 |
| DeepEdit* (Wang et al., 2024) | 13.37 | 8.23 | 59.85 | 45.38 |
| PokeMQA* (Gu et al., 2024b) | 15.77 | 7.64 | 63.97 | 37.76 |
| CHECK* (Simon & Ewetz, 2025) | 39.10 | 24.10 | 81.69 | 57.51 |
| QMEK (ours) | **49.47** | **38.68** | **83.94** | **62.88** |

curacy. Error analysis points towards the main failure point now being in the subquestion answering step, where the framework uses similarity thresholding to determine whether an edit is necessary for an intermediate reasoning step. This also seems to be a failure point across most frameworks, so work focusing on how to determine whether an edit is necessary in memory-based frameworks would be useful.

**Additional Models:** We provide framework question-answering accuries using older LLMs on MQuAKE-CounterFact (CF-3k) (Zhong et al., 2023) and MQuake-Temporal (T) (Zhong et al., 2023) in Table 5, and on MQuAKE-2002 (Wang et al., 2024) and MQuAKE-Hard (Wang et al., 2024) in Table 6. Frameworks are evaluated using GPT-J (Wang & Komatsuzaki, 2021), Vicuna (Chiang et al., 2023), and Falcon (Almazrouei et al., 2023). Across all datasets, models, and metrics, QMEK outperforms the other knowledge editing frameworks. QMEK achieves an average $14.05\%$ per-case accuracy increase and $16.04\%$ per-question accuracy increase over the next highest accuracy.

QMEK only relies on model knowledge for subquestion-answering, the point where model knowledge is necessary. The other frameworks rely on model knowledge and language processing capabilities for a range of tasks. Consequently, the ability of the frameworks to successfully complete tasks required for editing hinges on the underlying LLM's capabilities. Better models equate to better question-answering results, while worse models cause worse results due to failures in framework components. The models presented in Table 5 and Table 6 are older and less powerful, which we believe causes the fluctuations in the other frameworks accuracies. We believe QMEK's relatively stable performance across main paper and appendix evaluations can be attributed to the low reliance on causal language model language processing capabilities.

### C.2 KEBENCH

We provide the results of question-answering evaluations over the KEBench (Wei et al., 2024) dataset in Table 7. The frameworks are evaluated using the Qwen 2.5 (Yang et al., 2025), Mis-

Table 6: Per-case and per-question accuracy across the MQuAKE datasets. The top accuracy per column and per model is in bold. Results reported in CHECK are denoted with *.

| Dataset (Wang et al., 2024) | MQuAKE-2002 | | MQuAKE-Hard | |
|---|---|---|---|---|
| Accuracy Type (%) | Case | Question | Case | Question |
| Model | GPT-J (Wang & Komatsuzaki, 2021) | | | Size: 6B |
| GMeLLo-QA* (Chen et al., 2024) | 10.39 | 6.14 | 8.86 | 4.35 |
| MeLLo* (Zhong et al., 2023) | 17.18 | 8.13 | 6.76 | 2.64 |
| DeepEdit* (Wang et al., 2024) | 27.17 | 19.55 | 6.53 | 3.96 |
| PokeMQA* (Gu et al., 2024b) | 19.98 | 8.72 | 11.66 | 5.59 |
| CHECK* (Simon & Ewetz, 2025) | 56.59 | 40.86 | 35.90 | 23.85 |
| QMEK (ours) | **70.43** | **55.69** | **76.22** | **58.12** |
| Model | Vicuna (Chiang et al., 2023) | | | Size: 7B |
| GMeLLo-QA* (Chen et al., 2024) | 10.84 | 6.41 | 5.59 | 2.41 |
| MeLLo* (Zhong et al., 2023) | 9.84 | 5.13 | 1.86 | 0.85 |
| DeepEdit* (Wang et al., 2024) | 20.63 | 12.52 | 0.93 | 0.54 |
| PokeMQA* (Gu et al., 2024b) | 40.51 | 25.66 | 30.77 | 15.70 |
| CHECK* (Simon & Ewetz, 2025) | 63.74 | 41.99 | 48.72 | 29.68 |
| QMEK (ours) | **73.88** | **58.01** | **76.69** | **58.35** |
| Model | Falcon (Almazrouei et al., 2023) | | | Size: 7B |
| GMeLLo-QA* (Chen et al., 2024) | 6.50 | 3.63 | 5.36 | 3.34 |
| MeLLo* (Zhong et al., 2023) | 10.14 | 5.56 | 1.63 | 0.85 |
| DeepEdit* (Wang et al., 2024) | 19.53 | 12.02 | 2.80 | 1.24 |
| PokeMQA* (Gu et al., 2024b) | 19.93 | 9.14 | 13.05 | 7.46 |
| CHECK* (Simon & Ewetz, 2025) | 52.80 | 33.72 | 45.22 | 31.08 |
| QMEK (ours) | **70.93** | **56.26** | **76.22** | **57.58** |

Table 7: Per-question accuracy (%) on the KEBench dataset. The highest accuracy per column and per model is bolded.

| Dataset | KEBench (Wei et al., 2024) | | |
|---|---|---|---|
| Model | Qwen 2.5 | Mistral v0.3 | Llama 3.1 |
| PokeMQA | 34.27 | 40.71 | 33.38 |
| CHECK | 36.60 | 44.57 | 42.17 |
| QMEK | **45.43** | **52.11** | **56.43** |

tral v0.3 (Jiang et al., 2023), and Llama 3.1 (Grattafiori et al., 2024) LLMs. PokeMQA (Gu et al., 2024b) and CHECK (Simon & Ewetz, 2025) achieve an average 36.12% and 41.11% per-question accuracy, respectively. QMEK achieves an average 51.32% per-question accuracy, and an average 15.20% and 10.21% increase over PokeMQA and CHECK, respectively.

Additionally, we provided an evaluation of the decomposition accuracy of QMEK on KEBench in Figure 5 of the main paper. In this evaluation, QMEK achieved a 93% decomposition accuracy. This is a 15% higher decomposition accuracy than what QMEK achieved on MQuAKE-2002. However, we attribute this large jump in accuracy to KEBench only being 2-hop questions, while MQuAKE-2002 is composed 2-, 3-, and 4-hop questions. Questions with more hops are more difficult to decompose due to the larger error space.

### C.3 RIPPLE EDIT

To further illustrate the question decomposition capabilities of QMEK 's masked language modeling approach, we present another evaluation over unseen data in Figure 5. We evaluate the masked language model over the three Ripple Edit (Cohen et al., 2024) subsets, named popular, random,

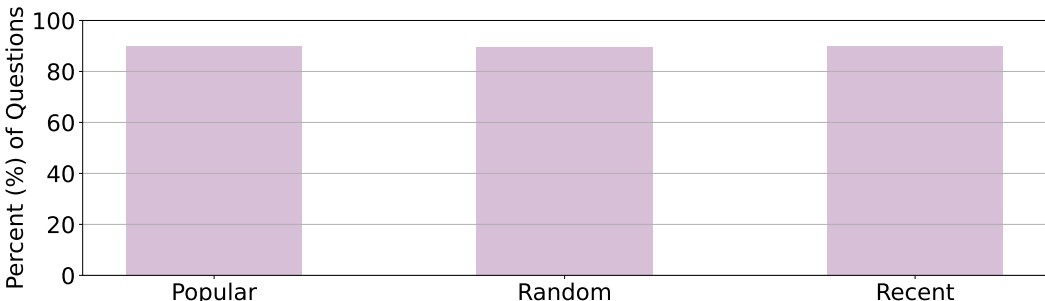

Figure 5: Question decomposition accuracy of the masked language model used by QMEK on the Ripple Edit subsets.

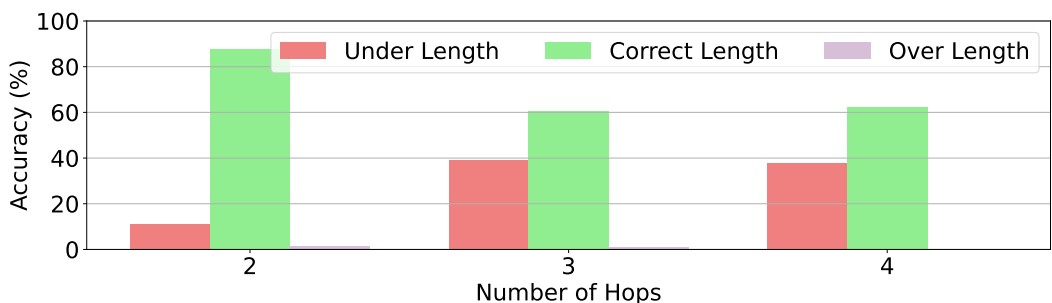

Figure 6: Question decomposition accuracy of QMEK's masked language model on the MQuAKE-2002 dataset, broken down by the number of question hops.

and recent. Each of the subsets is composed of only 2-hop questions. The relations that represent question hops in Ripple Edit share a low overlap with those used in MQuAKE, making it a suitable dataset to test model generalization to unseen relations. The masked language model is evaluated on its question decomposition accuracy.

QMEK 's masked language model achieves a decomposition accuracy of $89.8\%$, $89.2\%$, and $89.6\%$ on the popular, random, and recent subsets, respectively. Similar to the KEBench results, the decomposition accuracies are high because the model only had to decompose 2-hop questions. However, these results still further indicate that the masked language modeling approach for question decomposition generalizes well to unseen data.

Ideally, we could further evaluate the effectiveness of the masked language model by checking to ensure that the correct relations are extracted rather than just the correct number of relations. However, such an evaluation requires a large amount of human labeling that we do not currently have access to in order to augment existing datasets. A smaller scale evaluation is possible, but is heavily subject to the quirks of the small amount of data it would be over.

## D    ABLATION STUDIES

We present ablation studies for question decomposition accuracy in Section D.1, question-answering accuracy in Section D.2, and runtime in Section D.3. We also present a comparison of QMEK with and without a reranker in Section D.4.

### D.1    DECOMPOSITION ACCURACY ABLATION

We present an ablation study of the decomposition accuracy of QMEK's masked language model in Figure 6. The ablation study is over multi-hop questions with different numbers of question hops on the MQuAKE-2002 (Wang et al., 2024) dataset. This study measures whether the masked language model under extracts the number of question hops ($hops_{extracted} < hops_{ground\_truth}$), correctly

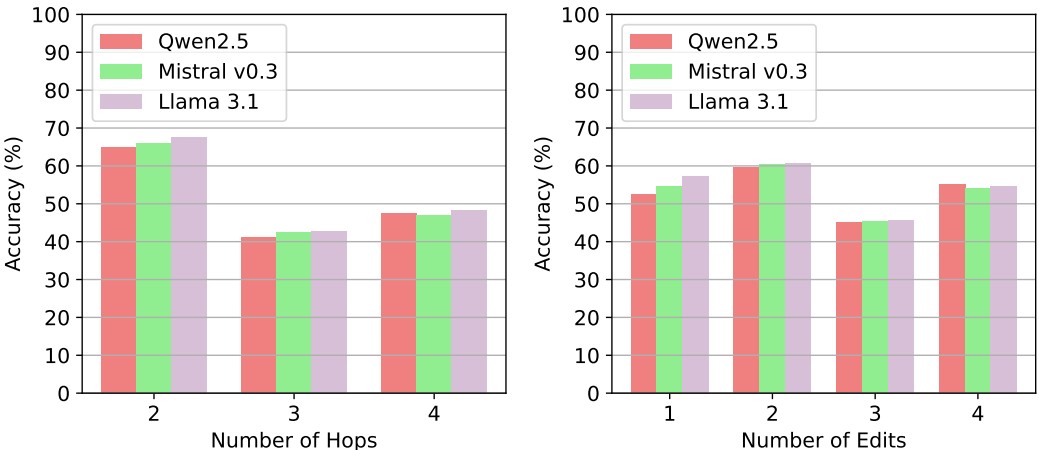

Figure 7: Per-question accuracy ablation over the number of question hops and edits per question on the MQuAKE-2002 dataset.

extracts the number of question hops ($hops_{extracted} == hops_{ground\_truth}$), or over extracts the number of question hops ($hops_{extracted} > hops_{ground\_truth}$). For example, the multi-hop question *Who is the wife of the creator of Microsoft?* has two question hops. Under extracting from this sentence is when only *wife* or *creator* are extracted. Correctly extracting is when both *wife* and *creator* are extracted. Over extracting from this sentence is when *wife*, *creator*, and 1 or more spans are extracted from the sentence.

QMEK correctly extracts 2 question hops from 88% of 2-hop questions, while extracting less than 2 hops in 11% of questions and more than 2 hops in 1% of questions. The number of correctly extracted question hops decreases for 3- and 4-hop questions, to 60% and 62%, respectively. Meanwhile, the number of under extracted question hops increases to 39% and 38% for 3- and 4-hop questions, respectively. At the same time, the number of over extracted question hops decreases to almost 0% for both numbers of question hops. Based on logs of the evaluations, we can see that this increase in under extracted question hops occurs because the masked language model labels two relations with the same question hops label. For example, given the question *Who is the wife of the creator of Microsoft?*, this error could manifest as the masked language model labeling *wife* and *creator* as belonging to the same relation. When they are extracted from the sentence, they will only be extracted as a singular relation rather than two separate relations, resulting in a decomposition error, and further negatively impacting overall question-answering performance.

### D.2 QUESTION-ANSWERING ACCURACY ABLATION

We present the per-question accuracy of QMEK on the MQuAKE-2002 dataset, broken down by the number of hops and number of edits per question, in Figure 7. Unsurprisingly, as the number of hops increases, question-answering accuracy decreases. More question hops means more places for framework components to fail. The main issues QMEK encounters that are amplified by increased question hops are correctly extracting relations and correctly determining when an edit is necessary. As discussed in Section D.1, the masked language model has an error where it erroneously labels two separate relations as a single relation. More question hops provides more opportunity for this error to occur. Additionally, determining when an edit is necessary is a currently under-addressed problem with vector embedding editors. QMEK determines whether an edit is necessary using a two-stage similarity thresholding method. While this approach is more robust than only using a cosine similarity threshold like other frameworks, the approach is still prone to error. The thresholds are set based on heuristics and do not provide any guarantee that necessary edits will always be similar enough to the current question hop to exceed both thresholds. As result, questions with greater numbers of question hops increase the overall error space.

It also makes sense the accuracies are relatively stable across different LLMs. QMEK does not offload any tasks onto the underlying LLM, only relying on the LLM for its knowledge during

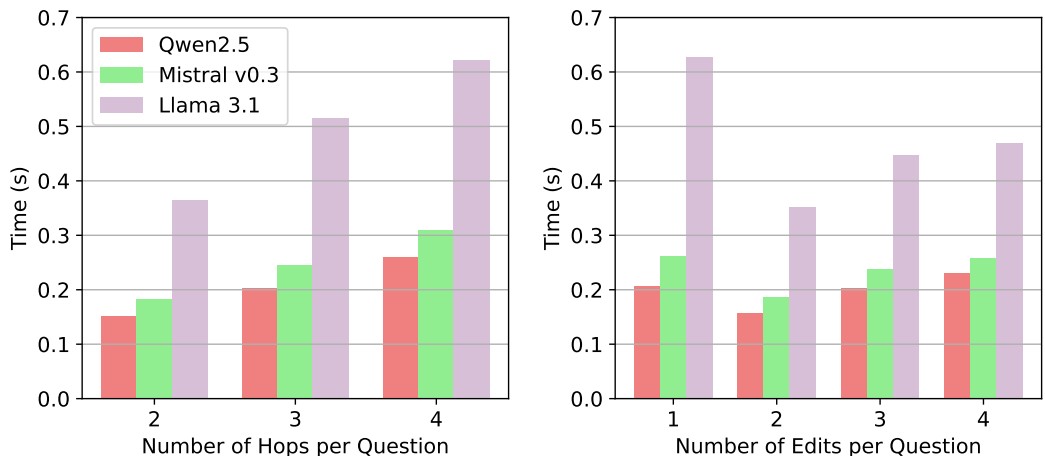

Figure 8: The average per-question runtime of QMEK on the MQuAKE-2002 dataset, broken down by number of hops and edits per question.

Table 8: Per-case and per-question accuracy across the MQuAKE datasets. Top accuracy per column and per model is bolded. QMEK with only a cosine similarity component (cos) and additionally a reranking component (rerank) are compared on question-answering accuracy.

| Dataset (Zhong et al., 2023) | MQuAKE-2002 | | MQuAKE-Hard | |
|---|---|---|---|---|
| Accuracy Type (%) | Case | Question | Case | Question |
| Model | Qwen 2.5 (Yang et al., 2025) | | | Size: 7B |
| QMEK - cos | 70.18 | 53.85 | 74.59 | 53.85 |
| QMEK - rerank | **73.48** | **57.59** | **76.92** | **58.35** |
| Model | Mistral v0.3 (Jiang et al., 2023) | | | Size: 7B |
| QMEK - cos | 70.98 | 54.71 | 74.59 | 54.00 |
| QMEK - rerank | **73.88** | **57.99** | **76.69** | **58.12** |
| Model | Llama 3.1 (Grattafiori et al., 2024) | | | Size: 8B |
| QMEK - cos | 71.78 | 55.71 | 74.83 | 54.23 |
| QMEK - rerank | **74.98** | **59.34** | **76.69** | **58.28** |

subquestion-answering. As a result, we would only expect to see minor accuracy fluctuations that occur based on differences in model knowledge, which is what occurs.

### D.3 RUNTIME ABLATION

We present the average per-question runtime of QMEK on the MQuAKE-2002 dataset, broken down by number of hops and edits per question, in Figure 8. The framework runtime increases as the number of question hops increase. This is expected behavior because more subquestion-answering iterations are required to answer an increased number of subquestions. However, even when processing 4-hop questions, QMEK only takes 0.26, 0.31, and 0.62 seconds when using Qwen 2.5 (Yang et al., 2025), Mistral v0.3 (Jiang et al., 2023), and Llama 3.1 (Grattafiori et al., 2024), respectively. These are incredibly short times when compared against other vector embedding frameworks.

The runtimes generally decrease when going from 1 to 4 edits. Querying the embedding space for relevant edits is less computationally expensive than prompting the underlying LLM to complete a triple. When more edits are necessary, QMEK uses edited answers more and queries the model less, resulting in decreased runtimes.

Table 9: Per-question runtime in seconds across the MQuAKE datasets. The highest accuracy per column and per model is bolded. QMEK with only a cosine similarity threshold (cos) and additionally a reranking threshold (rerank) are compared on runtime.

| Dataset | | MQuAKE-2002 | | | MQuAKE-Hard | | |
|---|---|---|---|---|---|---|---|
| Model | | Qwen 2.5 | Mistral v0.3 | Llama 3.1 | Qwen 2.5 | Mistral v0.3 | Llama 3.1 |
| QMEK - cos | | **0.15** | **0.19** | **0.44** | **0.19** | **0.21** | **0.48** |
| QMEK - rerank | | 0.20 | 0.23 | 0.46 | 0.23 | 0.27 | 0.46 |

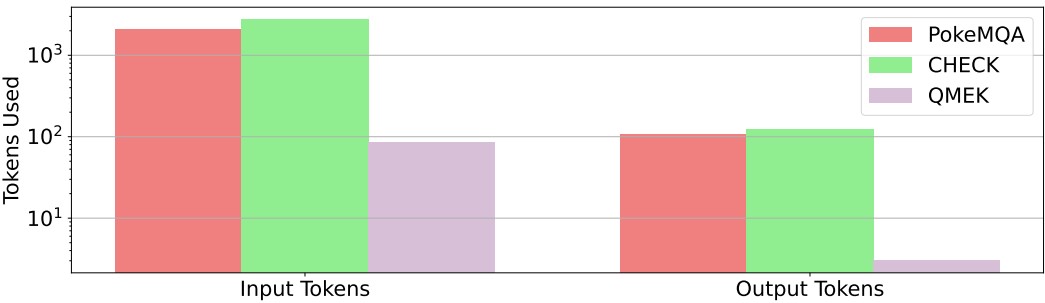

Figure 9: Editor average input and output token utilization of the underlying Qwen 2.5 model being edited on the MQuAKE-2002 dataset.

### D.4 RERANKING ABLATION STUDY

We provide an evaluation of QMEK's question-answering accuracy when only using a cosine similarity component (cos) and when using both a cosine similarity and reranking component (rerank). The evaluation is over the MQuAKE-2002 and MQuAKE-Hard datasets using the Qwen 2.5, Mistral v0.3, and Llama 3.1 LLMs. As expected, using a reranker increases the question-answering accuracy of QMEK. However, the accuracy increase is small, with an average $2.61\%$ per-case and $3.89\%$ per-question increase.

Additionally, a comparison of QMEK's runtime with only cosine similarity (cos) and with cosine similarity and reranking (rerank) is provided in Table 9. As expected, using only cosine similarity without reranking is faster. However, the runtime overhead of adding the reranker is only an average 0.03 seconds. Overall, using reranking gives a small boost to framework question-answering accuracy at a negligble time cost.

## E   LLM TOKEN UTILIZATION

To further explain the large gap in runtimes between QMEK and other editing frameworks, we present an evaluation on the average number of input and output tokens used by the underlying LLM at runtime in Figure 9. The evaluation is performed on MQuAKE-2002 using the Qwen 2.5 model. Larger token counts correspond with longer runtimes, so less tokens used points toward a more efficient framework. PokeMQA and CHECK use an average 2080 and 2766 input tokens and generate an average 106 and 121 new output tokens, respectively, per question. In contrast, QMEK uses an average of only $84$ input tokens and generates an average of only 3 new output tokens. That is a 1996 and 2766 input token difference and 103 and 118 output token difference from QMEK

PokeMQA relies heavily on the underlying LLM reasoning capabilities over large contexts, using the full editing context in each model call. CHECK prompts the model at different temperatures to find a suitable set of question hops, resulting in a large number of tokens used. Meanwhile, QMEK only prompts the model to complete a relational triple when model knowledge is required.

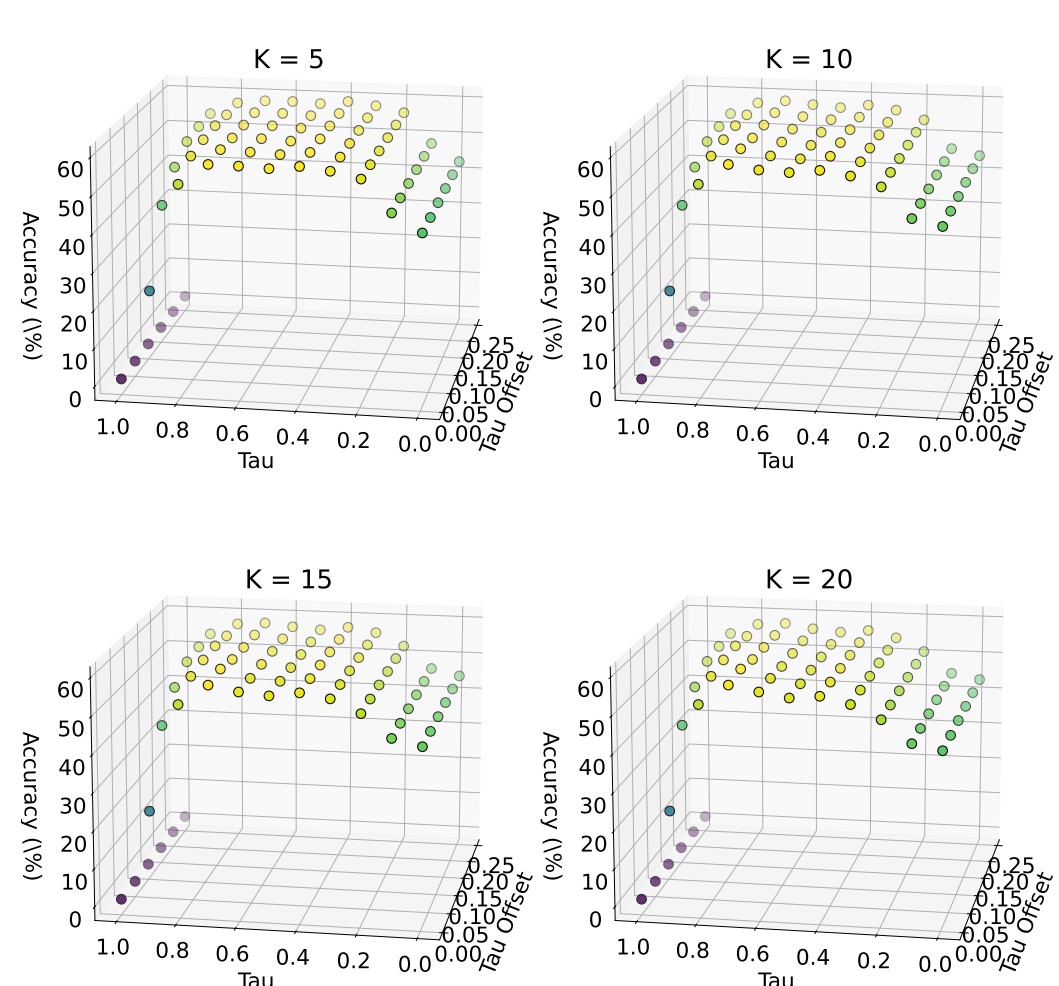

Figure 10: The parameter sweep using the **Qwen 2.5** large language model. The x-axis shows $\tau$ values on the range $[0.0, 1.0]$ with $0.1$ increments. The y-axis shows $\lambda$ values on the range $[0.0, 0.25]$ with $0.05$ increments. The z-axis shows the per-question accuracy over 900 questions from the MQuAKE-2002 dataset. Each graph shows a different top-$k$ value.

## F  THRESHOLDING PARAMETER SEARCH

The per-question accuracy of QMEK is evaluated over different values of $\tau$, $\lambda$, and $k$. We provide a parameter sweep over $\tau$ for values on the range $[0.0, 1.0]$ with $0.1$ increments, over $\lambda$ for values on the range $[0.0, 0.25]$ with $0.05$ increments, and over the top-$k$ values 5, 10, 15, and 20. For each of the $\tau$, $\lambda$, and top-$k$ value combinations, we evaluate QMEK over 100 2-, 3-, and 4-hop edit cases each, for a total of 300 unique edit cases and 900 multi-hop questions. We provide results of the parameter search on Qwen 2.5 in Figure 10, on Mistral v0.3 in Figure 11, and on Llama 3.1 in Figure 12. The x-axis shows the different $\tau$ values, the y-axis shows the different $\lambda$ values, and the z-axis shows the per-question accuracy associated with the $\tau$, $\lambda$, and top-$k$ combination. Each of the individual graphs correspond to a different top-$k$ value.

Across all 3 models, the per-question accuracy steadily rises from $\tau$ values of $0.0$ to $0.3$. Once $\tau$ reaches $0.3$, the per-question accuracy saturates through $\tau = 0.8$. The fluctuations in accuracy from changing top-$k$ and $\lambda$ values are generally within $5\%$, fluctuating between $58\%$ and $62\%$ accuracy. Per-question accuracies steeply decrease by after $\tau = 0.8$, with $\tau = 1.0$ uniformly achieving less

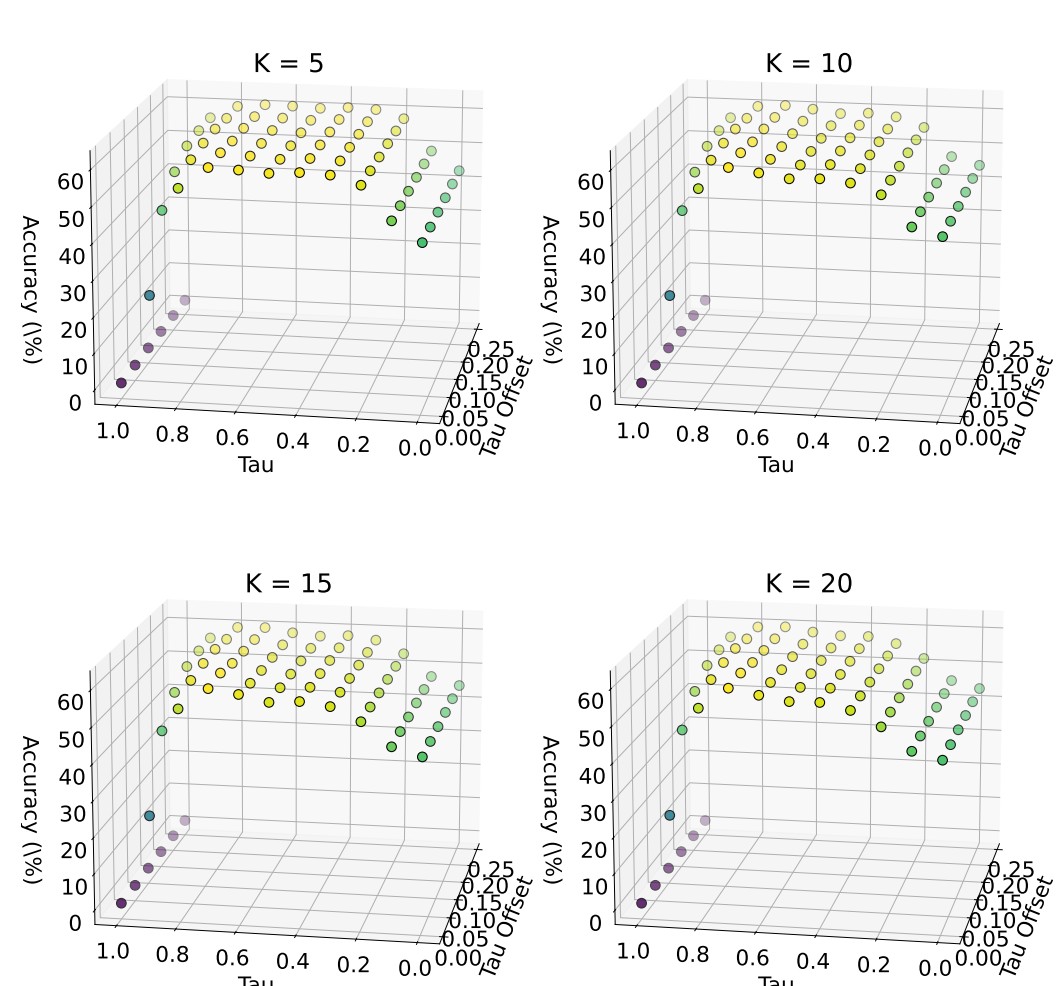

Figure 11: The parameter sweep using the **Mistral v0.3** large language model. The x-axis shows $\tau$ values on the range $[0.0, 1.0]$ with 0.1 increments. The y-axis shows $\lambda$ values on the range $[0.0, 0.25]$ with 0.05 increments. The z-axis shows the per-question accuracy over 900 questions from the MQuAKE-2002 dataset. Each graph shows a different top-$k$ value.

than $1\%$ per-question accuracy. Modifying $\lambda$ values has negligible impact on per-question accuracy when paired with $\tau$ values on the ranges of $[0.0, 0.6]$ and $1.0$. Conversely, sweeping through the whole range of $\lambda$ values produces accuracy fluctuations of no more than $3\%$ for $\tau$ values on the range of $[0.7, 0.8]$. However, there are large accuracy fluctuations caused by changing the $\lambda$ value when $\tau = 0.9$, but the per-question accuracies are around $20\%$ to $40\%$ while the accuracies on $\tau = [0.7, 0.8, 0.9]$ center around $55\% - 60\%$. Finally, the top-$k$ values can be seen to have similarly minimal effects on accuracy. Overall, the parameter with the largest impact is $\tau$.

The highest per-question accuracies achieved are 58.56, 60.78, and 62.11 for Qwen, Mistral, and Llama, respectively. The 58.56 accuracy is achieved on Qwen with the parameter pairs $(k = 5, \tau = 0.4, \lambda = [0.00, 0.05, 0.10, 0.15, 0.20, 0.25])$, $(k = 10, \tau = 0.7, \lambda = 0.05)$, and $(k = 15, \tau = 0.7, \lambda = 0.05)$. The 60.78 accuracy is achieved on Mistral with the parameter pairs $(k = [10, 15], \tau = 0.7, \lambda = 0.05)$. The 62.11 accuracy is achieved for Llama with the parameter combination $(k = 15, \tau = 0.7, \lambda = 0.05)$. The results on Qwen with $k = 5$ can be seen as an anomoly occurring due to a small sample set. However, all models have the parameter combination $(k = 15, \tau = 0.7, \lambda = 0.05)$. In our evaluations, we choose to use $\tau = 0.7$ and $\lambda = 0.05$.

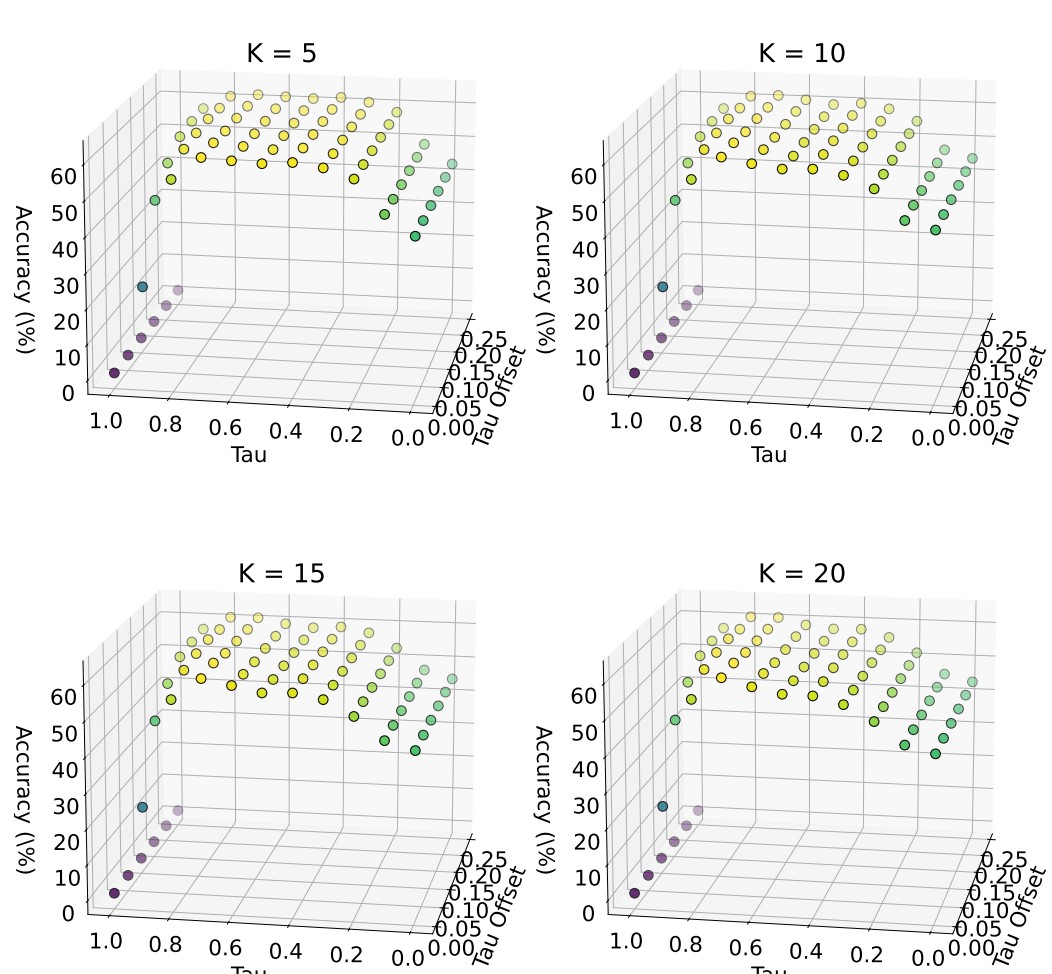

Figure 12: The parameter sweep using the **Llama 3.1** large language model. The x-axis shows $\tau$ values on the range $[0.0, 1.0]$ with $0.1$ increments. The y-axis shows $\lambda$ values on the range $[0.0, 0.25]$ with $0.05$ increments. The z-axis shows the per-question accuracy over 900 questions from the MQuAKE-2002 dataset. Each graph shows a different top-$k$ value.

However, we choose to use $k = 20$ instead of $k = 15$. Qwen, Mistral, and Llama achieve a $58.44\%$, $60.44\%$, and $61.78\%$ per-question accuracy, respectively, with the parameter combination $(k = 20, \tau = 0.7, \lambda = 0.05)$. These are only a $0.12\%$, $0.34\%$, and $0.33\%$ difference from the top accuracies, so the difference is negligible. In theory, a larger number of edits passed to the cross-encoder should result in better reranking because the cross-encoder ranking mechanism provides better similarity scores than cosine similarity. We attribute this minor deviation again to a small sample set size. The final parameter combination we use is $(k = 20, \tau = 0.7, \lambda = 0.05)$.

## G  DATASET GENERATION

The masked language model used by QMEK is trained using a dataset that is created by extracting information from existing knowledge editing datasets. In this paper, we use information extracted from MQuAKE-CounterFact-3k v2 (Zhong et al., 2023), but this approach can be used on any knowledge editing dataset that provides each question hop as a relation.

```
Provide the object for the following triple: | Paris | capital |
Object: France|

Provide the object for the following triple: | Joe Biden | spouse |
Object: Jill Biden|

Provide the object for the following triple: | Hatchet | author |
Object: Gary Paulson|

Provide the object for the following triple: | Rainn Wilson | country of citizenship |
Object: United States|

Provide the object for the following triple: | iPhone5 | manufactured by |
Object: Apple|
```

Figure 13: The in-context learning prompt provided to the LLM for relational triple completion.

First, we extract all multi-hop questions and ground truth relations that correspond to their question hops from the knowledge editing dataset. These categories are explicitly given in the MQuAKE datasets. The questions are paired with their corresponding question hop relations. Next, we match the relation spans to spans in the multi-hop questions to get the multi-hop question string indices corresponding to the relations. We also manually create relation aliases for span matching because the ground truth relations are not always directly represented in the multi-hop question. For example, the ground truth relation *country of citizenship* can appear in the multi-hop question as *citizen of*, warranting the addition of *citizen of* as a relation alias.

Each word in the question relations are marked with a start and stop index to denote the true position of the hop in the multi-hop question. The indices are used to map the words composing each relation to tokens during training. This further allows for the creation of token labels for training. We also separate the words by relation hop they belong to so that the masked language model can be trained to differentiate between different relations instead of a binary *is* or *is not* a question hop relation. We discard any multi-hop questions that we were unable to locate the string position of at least 1 question hop relation. Each data point in the masked language model training set contains a multi-hop question, the ground truth relations corresponding to question hops, and the indices of each word that maps to a question hop.

## H  RELATIONAL TRIPLE COMPLETION

QMEK answers subquestions using LLM knowledge when no edit is necessary. QMEK represents subquestions as incomplete relational triples $t = (s, r, o^u)$, where the incompleteness comes from the object being unknown $o^u$. The LLM is prompted with a relational triple completion task to find the unknown object $o^u$, completing the triple. LLMs are adept at few-shot learning (Dong et al., 2024) and link prediction (Alqaaidi & Kochut, 2024; Wei et al., 2023), making this approach a good choice for answering subquestions with LLM knowledge. Keeping subquestions as relational triples also eliminates any chance for errors that occur when translating triples to natural language.

The few-shot learning prompt used for triple completion is provided in Figure 13. Triple completion can be concretely described using the first example in the prompt. The subject and relation given are *Paris* and *capital*. The expected answer to complete the triple is *France*, because *Paris is the capital of France*. If this example were passed to the model, it would use the context of *Paris* and *capital* to complete the triple with *France*.

