# OpenReview forum: "Question Decomposition using Masked Language Modeling for Knowledge Editing"
_ICLR.cc/2026/Conference — ICLR 2026 Conference Withdrawn Submission_

### Official Review · Reviewer_vQ1q · 2025-10-27

**Soundness:** 2
**Presentation:** 3
**Contribution:** 2
**Rating:** 4
**Confidence:** 4

**Summary:**

This paper addresses a key bottleneck in knowledge editing for multi-hop question-answering (MQA): the unreliability and high latency of using large causal language models (CLMs) for question decomposition. The authors propose QMEK (Question Decomposition using Masked Language Modeling for Editing Knowledge), a new framework that reformulates decomposition as an extractive task rather than a generative one. Instead of prompting a large CLM to generate subquestions, QMEK uses a lightweight masked language model (MLM) to perform token classification, directly identifying and extracting relation triples from the input question. This decomposition is paired with a subquestion answering module that operates entirely in the relational triple space, querying an edit bank or using the base LLM for triplet completion, thus avoiding error-prone translations between natural language and structured triples.

**Strengths:**

1. The authors correctly identify that existing CLM-based approaches (both NL and triplet-generating) are error-prone and slow.  Reframing this as a token classification task (as visualized in Figure 1) is an elegant, technically sound, and well-motivated solution to this specific bottleneck.

2. Achieving a simultaneous 10.2x speedup and a 17.5% accuracy gain is a rare and significant achievement. The dramatic reduction in runtime (e.g., 0.31s for QMEK vs. 3.30s for CHECK ) is a major practical contribution, moving the technology from a latency that is unacceptable for many real-time applications to one that is viable.

3. The paper is exceptionally well-written and organized.

**Weaknesses:**

1. The paper's core innovation—the MLM decomposer—is presented as an extractive token classifier. This approach seems highly dependent on the presence of exact relation strings (e.g., "birthplace", "spouse", "director" from Figure 1) in the input question. It is unclear how this method handles linguistic variation and paraphrasing.

2. The QMEK framework appears to assume that all multi-hop questions can be neatly decomposed into a linear chain of relational triples (s, r, o) -> (o, r', o') .... This assumption holds for the MQuAKE datasets, which are synthetically constructed from knowledge graph paths. However, many real-world compositional questions are not simple chains.

3.Regarding the MLM decomposer: How robust is it to paraphrases of relations? For example, if the training data used "spouse of" (as in Figure 1), could it correctly identify "partner of" or "husband of" at inference time?

4. While adequate for contextualizing the baselines, is missing recent work like [1] [2] [3].

[1] MQuAKE-Remastered: Multi-Hop Knowledge Editing Can Only Be Advanced With Reliable Evaluations. Zhong et al ICLR 2025.

[2] Decoding by Contrasting Knowledge: Enhancing LLMs' Confidence on Edited Facts. Bi et al ACL 2025.

[3] AlphaEdit: Null-Space Constrained Knowledge Editing for Language Models. Fang et al ICLR 2025.

**Questions:**

See above.

---

### Official Review · Reviewer_kDvj · 2025-10-31

**Soundness:** 2
**Presentation:** 3
**Contribution:** 2
**Rating:** 4
**Confidence:** 3

**Summary:**

This paper introduces QMEK (Question Decomposition using Masked Language Modeling for Editing Knowledge), a novel framework designed to improve knowledge editing in large language models (LLMs) for multi-hop question answering (MQA). Unlike existing approaches that rely on causal language modeling (CLM) for question decomposition, often leading to hallucinations and high inference costs, QMEK reformulates the decomposition process as a masked language modeling (MLM) task. Overall, the paper’s contributions lie in reformulating question decomposition for knowledge editing as a masked modeling task.

**Strengths:**

The paper introduces a novel perspective on knowledge editing for multi-hop question answering by reformulating question decomposition as a masked language modeling (MLM) task rather than the conventional causal language modeling (CLM) approach. The reported experimental improvements, an average of 17.5% accuracy increase and 10.2× speedup over five state-of-the-art baselines across three datasets.

**Weaknesses:**

1. The evaluation appears to be constrained to four relatively simple datasets, with a maximum hop count of four. Moreover, three of these datasets seem to be closely related variants of the same benchmark, which limits the diversity of the evaluation. This makes it difficult to fully assess the robustness and general applicability of QMEK. To strengthen the empirical validation, the authors are encouraged to include additional datasets, ideally those with more complex multi-hop structures, and to test against a wider range of strong baselines that employ causal language models (CLMs) strategies. Such an expansion would provide more convincing evidence for the claimed performance gains.

2. The proposed approach relies on extracting relational triples from the question and then inferring missing entities based on existing relational context. While this design is elegant, it introduces a structural limitation: when an entire triple (subject, relation, object) is missing or misidentified, the current algorithm appears unable to recover, leading to cascading failures in multi-hop reasoning. This brittleness may restrict the method’s generalizability to real-world questions that contain implicit or underspecified relations. The paper would benefit from (a) an analysis of the frequency and impact of such failure cases, (b) proposed remedies(e.g., the use of external retrieval for missing triples).

**Questions:**

The paper mentions that QMEK infers missing entities within existing triples but does not clarify how it handles cases where the entire triple is absent or incorrectly extracted. Could the authors elaborate on whether any fallback mechanisms (e.g., probabilistic completion, retrieval augmentation, or semantic similarity matching) were explored? Have you quantified the proportion of decomposition failures attributable to missing triples, and if so, how do they impact downstream MQA accuracy?

The datasets used for evaluation seem to have relatively low hop complexity (≤4) and overlapping characteristics. Could the authors justify their dataset choice more clearly?

---

### Official Review · Reviewer_JwG3 · 2025-11-01

**Soundness:** 3
**Presentation:** 3
**Contribution:** 2
**Rating:** 4
**Confidence:** 4

**Summary:**

This paper proposes QMEK, a knowledge editing framework for multi-hop question-answering that uses masked language modeling (MLM) for question decomposition instead of the causal language modeling (CLM) approach used by existing methods. The core thesis is that reformulating question decomposition as a token classification task (identifying which words belong to which relation) reduces hallucinations and improves both accuracy and speed compared to iterative token generation with CLMs. The authors report substantial improvements: 17.5% higher accuracy and 10.2× speedup over state-of-the-art baselines across multiple datasets.

**Strengths:**

1. The core idea that extracting relations from existing text is easier than generating them is intuitive and well-motivated.
2. The unified triplet representation offers a clean and practical framework for implementation.
3. The ablation studies and appendices reflect rigorous and detailed validation.
4. The error analysis is transparent, especially in acknowledging the performance drop on 3–4 hop questions.

**Weaknesses:**

1. The core claim that MLM outperforms CLM for question decomposition is not well supported. The experiments compare a fine-tuned BERT with zero/few-shot CLMs (PokeMQA, CHECK), conflating model type with training setup. A fair test requires fine-tuned GPT/Llama baselines on the same dataset.
2. The model’s performance drops sharply on harder (3–4 hop) questions and external datasets. The assumption that relations are contiguous text spans is unrealistic and unverified, limiting scalability to complex reasoning cases. Hyperparameters are tuned on MQuAKE data and highly sensitive. No evidence shows they generalize to other datasets, suggesting overfitting to the evaluation set.
3. The reported speedup mainly comes from fewer LLM calls, not algorithmic novelty. Baselines could achieve similar gains via caching or lightweight modules.

**Questions:**

1. I strongly recommend adding a critical missing experiment: fine-tune a comparable-sized CLM (e.g., GPT-2 or a small Llama) on the same question decomposition dataset used for BERT. This would directly test whether the observed gains arise from the MLM paradigm itself or simply from fine-tuning effects. It’s possible that a fine-tuned CLM, given enough supervision, could achieve similar or even superior performance due to its stronger generative capabilities.

2. In addition, while the paper contrasts its approach with existing knowledge editing frameworks, it largely overlooks the broader question decomposition and semantic parsing literature. Prior work in relation extraction and semantic parsing could provide more meaningful baselines than prompted LLMs. A discussion or comparison with these methods would help clarify what is genuinely novel about this approach and where it fits within the existing landscape.

---

### Official Review · Reviewer_Xunh · 2025-11-01

**Soundness:** 2
**Presentation:** 3
**Contribution:** 2
**Rating:** 4
**Confidence:** 4

**Summary:**

This paper addresses knowledge editing for multi-hop question answering (MQA), aiming to modify the internal knowledge of language models without retraining while ensuring they correctly use the edited knowledge during multi-hop reasoning. The authors propose QMEK, a framework that uses an entity linking model to identify question subjects and a masked language model (BERT) to label relation words in questions. Multi-hop questions are decomposed into incomplete triples, which are then completed using either an edit memory bank or the LLM itself.

**Strengths:**

- The approach of reformulating question decomposition from causal language modeling (text generation) to masked language modeling is novel. Using traditional NLP techniques like entity linking and token/span classification to build a multi-hop QA pipeline is a creative direction.
-  The paper presents extensive experiments across multiple datasets and models, demonstrating consistent improvements over baselines. The evaluation covers multiple dimensions including accuracy, decomposition accuracy, and runtime, providing a thorough empirical analysis.
- The proposed method shows significant improvements in both speed and accuracy compared to baseline approaches, which could be valuable in specific deployment scenarios.

**Weaknesses:**

- My primary concern is whether this complex approach is necessary. The core task (answering questions when LLMs encounter knowledge not seen during training) has been extensively studied in recent work on LLM-centric RAG, agentic question answering, and various chain-of-thought methods. Modern LLMs can already perform question decomposition through structured prompting (e.g., structured chain-of-thought with iterative filling). The proposed method appears over-engineered, requiring entity linking, MLM fine-tuning, embedding models, and cross-encoders, which increases system complexity and limits generalizability. Critically, the paper lacks comparison with simple baselines where LLMs directly perform structured reasoning through agentic question decomposition.
- The method's robustness across diverse domains is questionable. Each component (BERT model, entity linker, embedding model, cross-encoder) requires domain-specific data or models. This creates significant domain transfer challenges. Entity linking remains unsolved in many domains. For example, linking medical entity mentions to UMLS concepts often yields poor performance. Since the paper only evaluates on general knowledge domains, testing on domain-specific datasets is necessary. Additionally, analysis of how entity linking errors impact overall performance would strengthen the evaluation.
-  Many types of knowledge cannot be perfectly represented as structured triples, including temporal information, numerical relations, complex logical relations, and implicit causal relationships. Furthermore, the method assumes questions always contain explicit relation words, but many questions involve implicit relations or expression styles without clear relation markers.
- The paper does not provide statistical significance tests for the reported improvements.

**Questions:**

- Can you provide a direct comparison with a simple baseline where an LLM is asked to perform structured multi-hop reasoning in a zero-shot or few-shot setting? Specifically, how does your method compare to modern LLMs that perform question decomposition through structured prompting techniques?
- What is the deployment cost trade-off? How do you justify the complexity of fine-tuning MLM, maintaining entity linkers, embedding models, and cross-encoders versus using a slightly slower but much simpler LLM-based approach?

---

### Note · Authors · 2025-11-12

**Comment:**

Thank you to the reviewers for taking the time to review our paper.

**Withdrawal Confirmation:**

I have read and agree with the venue's withdrawal policy on behalf of myself and my co-authors.